# Feasibility of Provision and Vaccine Hesitancy at a Central Hospital COVID-19 Vaccination Site in South Africa after Four Waves of the Pandemic

**DOI:** 10.3390/diseases12060113

**Published:** 2024-05-24

**Authors:** Shanal Nair, Khanyisile Tshabalala, Nevilene Slingers, Lieve Vanleeuw, Debashis Basu, Fareed Abdullah

**Affiliations:** 1Steve Biko Academic Hospital, Pretoria 0001, South Africa; khanyisile.tshabalala@up.ac.za (K.T.); debashis.basu@up.ac.za (D.B.); fareed.abdullah@mrc.ac.za (F.A.); 2Department of Public Health Medicine, School of Health Systems and Public Health, Faculty of Health Sciences, University of Pretoria, Pretoria 0001, South Africa; 3Office of AIDS and TB Research, South African Medical Research Council, Pretoria 0001, South Africa; nevilene.slingers@mrc.ac.za (N.S.); lieve.vanleeuw@gmail.com (L.V.); 4Division of Infectious Diseases, Department of Internal Medicine, Faculty of Health Sciences, University of Pretoria, Pretoria 0001, South Africa

**Keywords:** vaccination, vaccine uptake, COVID-19, hospitals, vaccine hesitancy

## Abstract

Background: As mortality declined significantly during the fourth and fifth waves compared to previous waves, the question of the future role of COVID-19 vaccination arose among both experts and the public in South Africa. Turning attention away from the general public, now considered to be at very low risk of severe COVID-19 disease, a commonly held view was that the vaccination campaign should focus only on those who remain highly vulnerable to severe disease and death from COVID-19. Primary amongst this group are patients with common chronic diseases attending hospital outpatient departments. We hypothesized that providing COVID-19 vaccinations on-site at a central hospital will increase uptake for the patients with co-morbid chronic conditions who need them most in the Omicron phase of the pandemic. Aim: Evaluate the acceptability, need, and uptake of a hospital-based vaccination site for patients attending the medical hospital outpatient departments. Objectives: To assess vaccination uptake, coverage, and hesitancy in people attending a central hospital, to determine factors associated with and influencing vaccination uptake, and to document implementation and assess acceptability of the vaccination project among staff and persons attending the hospital. Methods: Mixed-methods study using quantitative and qualitative methods. Results: Of the 317 participants enrolled in the study, 229 (72%) had already received at least one dose of the COVID-19 vaccine. A total of 296 participants were eligible for a first vaccination, additional vaccination, or booster vaccination according to the South African Department of Health guidelines. Of those previously vaccinated, 65% opted for an additional dose on the day it was offered (same day). Only 13 previously unvaccinated participants (15% of vaccine naïve participants) opted for vaccination, increasing vaccine coverage with at least one dose from 72% to 76%. Approximately 24% (*n* = 75) of all participants refused vaccination (vaccine hesitant). Variables tested for an association with vaccination status demonstrated that age reached statistical significance. Emerging themes in the qualitative analysis included perceptions of vulnerability, vaccine safety and efficacy concerns, information gaps regarding vaccinations, the value of convenience in the decision to vaccinate, and the role of health promoters. Conclusions: This study has shown that it is logistically acceptable to provide a vaccination site at a large hospital targeting patients attending outpatient services for chronic medical conditions. This service also benefits accompanying persons and hospital staff. Access and convenience of the vaccination site influence decision-making, increasing the opportunity to vaccinate. However, vaccine hesitancy is widespread with just under one-quarter of all those offered vaccinations remaining unvaccinated. Strengthening health education and patient–clinician engagement about the benefits of vaccination is essential to reach highly vulnerable populations routinely attending hospital outpatient departments with an appropriate vaccination program.

## 1. Introduction

South Africa has recorded just over 4 million COVID-19 cases and more than 102,000 deaths attributable to the illness as of February 2024 [1]. South Africa has now entered the recovery stage of the COVID-19 pandemic, and the majority of South Africans have either natural immunity from prior COVID-19 infection or vaccine-induced immunity [2,3]. New COVID-19 infection, or re-infection, is more likely to cause severe disease in highly vulnerable groups. These groups include the elderly, those with concurrent infections such as TB and HIV, autoimmune diseases, and those with co-morbidities such as diabetes, hypertension, chronic cardiac disease, chronic kidney disease, and malignancy [4,5,6]. Within 12 months of the onset of the pandemic, vaccines emerged as the most important medical countermeasure to prevent severe disease and mortality secondary to COVID-19 [7].

South Africa’s COVID-19 vaccination campaign commenced on 17 February 2021, with the rollout conducted in three phases, in coordination with provincial health departments and the private healthcare sector. The first phase focused on vaccinating front-line healthcare workers. Access to vaccines in this phase was through work-based vaccination programs at district-level private and public hospitals, outreach work-based vaccination programs through mobile teams going to health facilities, and vaccination centres at remote or facility-based vaccination units [7]. The second phase of the vaccination rollout, which commenced on 17 May 2021, focused on essential workers and high-risk groups, followed by the third phase, which commenced in September 2021, reaching the wider population. Access to vaccines in these phases, phases two and three, were obtained through public facility vaccination sites (primary healthcare clinics), vaccination centres (community pharmacies, general practitioners, or non-profit organizations), outreach vaccination programs (mobile clinics), and work-based vaccination programs [7]. Vaccinations provided as part of the national rollout included the Pfizer vaccine, of which two doses were required for a person to be fully vaccinated, and the Janssen vaccine (J&J), which as per the WHO, recommended two doses of the J&J vaccine to achieve higher efficacy and endpoints [8].

By July 2022, when this vaccination study commenced, South Africa had already experienced its fourth wave and reached the end of its fifth wave, both of which manifested a decreased clinical severity and a decoupling of infections from mortality [9]. Government spokespersons and experts concluded that the worst of the pandemic was over and non-pharmaceutical restrictions, including mask mandates, were rapidly being reversed. Although governments continued a policy of vaccinations and boosters, the uptake of COVID-19 vaccinations ground down to very low levels in this period reflecting a drastically diminished risk perception in the general public. The study investigators took the view that the vaccination programme needs to focus on hospital outpatient departments which were being attended by the population of chronically ill persons at highest risk of severe disease from future infections. As it became clearer that COVID-19 vaccines had little benefit against vaccine-evading Omicron variants, the goal of the public health response to prevent infection through vaccination largely fell away, and only the elderly or highly vulnerable were being advised to have boosters [10].

Being elderly and having comorbidities such as diabetes, hypertension, chronic kidney disease, autoimmune conditions, and cancer are associated with severe forms of COVID-19 and high mortality [5]. The South African population has a high burden of non-communicable diseases (NCDs), with 50.9% of all deaths in South Africa attributed to NCDs [6]. This can be further broken down into 18.9% of deaths secondary to cardio- and cerebrovascular disease and 7.7% of all deaths due to diabetes and chronic kidney disease, with an estimated 4.5 million South Africans living with diabetes [6,11]. The country also experiences a very high burden of HIV, with an estimated HIV prevalence rate of 13.5% of the population, equating to approximately 8 million people living with HIV [12]. HIV infection has been associated with an increased risk of severe COVID-19 disease and mortality [13]. Not only is the country grappling with HIV, but there is a concurrently large TB burden, with a prevalence of 852 cases per 100,000 people documented between 2017 and 2019 [12]. There is also a dual burden of TB and HIV co-infection, with 53% of TB patients also living with HIV [12]. Both living with HIV and having current TB infection have been independently associated with increased COVID-19 mortality [14].

Given the ongoing COVID-19 vulnerabilities among people with NCDs, HIV, and TB, these highly vulnerable groups are likely to benefit from vaccinations for COVID-19. As of February 2024, South Africa has vaccinated 22.8 million individuals with at least one dose of the vaccine, which is just under half of the adult population in the country [15].

However, vaccine hesitancy should not be overlooked [16], which is defined as “a delay in acceptance or refusal of vaccination despite availability of vaccination services [17]”. Common findings amongst other studies found reasons for vaccine hesitancy include concerns over vaccine safety and efficacy, side effect profile, rapidity in the development of the COVID-19 vaccine, and a decreased risk perception of acquiring COVID-19 [18,19,20,21,22].

Notwithstanding this, easy access to convenient vaccination delivery is likely to play an important role in the decision to vaccinate. People with NCDs, HIV, and TB visit hospitals regularly for routine outpatient visits and collection of medication, and therefore can easily be offered a vaccination during their hospital visits. This project aimed to test the acceptability, need, and uptake of COVID-19 vaccinations as part of the outpatient services at the Steve Biko Academic Hospital in Tshwane.

## 2. Aim and Objectives

### 2.1. Aim

To evaluate the acceptability, need, and uptake of a hospital-based vaccination service for patients attending the hospital’s medical outpatient departments.

### 2.2. Objectives

To assess vaccination uptake, coverage, and hesitancy among people attending a central hospital.To determine factors associated with and influencing vaccination status.To document the implementation and assess the acceptability of the vaccination project among staff and persons attending the hospital.

## 3. Materials and Methods

### 3.1. Study Design

The study is a mixed-methods study using both quantitative and qualitative data collection methods.

### 3.2. Setting

The study was conducted at the Steve Biko Academic Hospital (SBAH). SBAH is an 845-bed academic hospital in Pretoria South Africa. The hospital delivers specialized medical services for the Tshwane district, as well as for patients from around the province and country. SBAH caters to a population of just over 4 million people, and further sees on average 2000 patients in the outpatient clinics monthly. SBAH, in collaboration with the University of Pretoria, was the initiating site for the COVID-19 vaccination program in the country. The vaccination process at this site continued until the end of the COVID-19 pandemic, declared by the World Health Organization on 5 May 2023 [23]. SBAH, being the academic hospital for the university, was fit to continue the vaccination provision. This site was selected due to its centrality, regional importance, and the location of the researchers. The vaccination service was offered to patients attending the medical outpatient departments. The vaccination site employed health promoters to assist in increasing awareness and knowledge of the vaccine and the vaccination site. Patients presenting to the vaccination site were offered coffee or tea and a muffin, and their prescriptions were expedited at the pharmacy to mitigate the time spent by participants attending the vaccination site.

### 3.3. Sample Population, Size, and Sampling

#### 3.3.1. Enrolment for the Quantitative Component

All persons visiting the hospital were invited to participate in the study. The only eligibility criteria required for the study were participants to be 18 years of age or older. Those eligible for COVID-19 vaccination, according to the South African Department of Health protocol, were invited to receive a COVID-19 vaccine. An informed consent document was developed and approved by the institutional ethics committee. Using this document, consent was obtained from study participants by well-trained research investigators. All participants were counselled on the aims, objectives, methods, the participant’s role in the study, and the confidentiality clause that described how the data will be used and protected.

The minimum sample size required was calculated to include 234 participants to achieve a confidence interval of 95% and a 5% margin of error.

#### 3.3.2. Enrolment for the Qualitative Component

A convenience sample of 30 participants was drawn from people who chose to vaccinate (*n* = 15) and people visiting the hospital but opted not to vaccinate (*n* = 15). In addition, staff members involved with the vaccination site operations were invited to participate in the in-depth interviews. All participants included in the in-depth interviews were 18 years of age or older. Consent from participants was obtained as described above. Only participants enrolled for the qualitative component were reimbursed in the form of a monetary voucher to a local supermarket. This was approved by the institutional ethics board.

### 3.4. Data Collection

The questions for the questionnaire were developed to answer the aims and objectives of the study, taking into consideration existing literature and expert opinion [22,24,25].

A pilot phase was conducted with staff and patients before data collection. Individuals were randomly selected to determine the questionnaire’s comprehensibility, wording, and clarity. These records were not included in the final analysis.

The quantitative data were collected using a self-administered or interviewer-assisted questionnaire on a tablet, directly onto REDCap (Research Electronic Data Capture) a secure, web-based application designed to support data capture for research studies. The tool collected information on socio-demographic characteristics (nationality, age, gender, and employment status), medical history including comorbidities and concurrent infections (diabetes, hypertension, HIV, and other diseases), COVID-19 vaccination history (dates of previous vaccinations and the uptake of a booster), and the current desire to vaccinate on the day of the hospital visit, including reasons for not taking up a same-day vaccine. The questionnaire was comprised of close-ended questions. Participants were then invited to vaccinate on-site on the day of their visit.

The qualitative data were collected by in-depth interviews conducted with the team that operates the vaccination site, the hospital staff and management, and the patients, including those vaccinating and those not vaccinating. Interviews were conducted in a private space and were recorded on a voice recorder to be transcribed at a later stage.

Data collection for the quantitative data commenced in July 2022 and continued until the end of March 2023. The in-depth interviews commenced in December 2022 and continued until the end of February 2023.

### 3.5. Data Analysis

#### 3.5.1. Quantitative Data

Data were exported into STATA 17 for analysis. We calculated Cronbach’s alpha coefficient on the questionnaire to determine reliability and coherence between the items. The overall Cronbach alpha score for reliability for the questionnaire was 0.6. This is of an acceptable level for internal consistency.

Demographic and clinical data of all participants enrolled in the study were summarized using descriptive statistics. Continuous variables were summarized into mean, minimum, maximum, and standard deviations. Categorical variables were summarized using frequencies, counts, and standard deviations. Vaccination coverage, uptake, and hesitancy were described, with vaccine hesitancy being measured by the proportion of participants that remain vaccine naïve at the end of the study. The number and the proportion of those vaccinated were reported and stratified by age, gender, employment status, nationality, and co-morbidity status. Tests of association performed included the chi^2^ test. A univariable followed by a multivariable analysis was conducted to assess the association of variables to the outcome of vaccination status. All statistical tests were conducted at 5% significance levels.

#### 3.5.2. Qualitative Data

All qualitative data were analysed inductively using Graneheim and Lundman manifest and latent content analysis methods [26]. Data were transcribed in English and checked against the original recording by the researchers to ensure accuracy. Following each interview, field notes were written to capture the context and non-verbal communication. Following each day of interviews, the investigators discussed the interviews and field notes, noting initial thoughts and meanings, and emerging themes related to the original research question. Emerging themes and conclusions from these discussions were used to guide further interviews where appropriate. Together with the co-investigators, the lead investigator read through each of the transcripts, noted initial thoughts, and began coding the data. Initial codes were grouped into categories that were then further transformed into major themes. The investigators discussed the first phase of the analysis, and once there was consensus, the codes, categories, and themes generated from this exploration were used in the analysis of the remaining transcripts; thus, the analysis of the qualitative data was an iterative process.

## 4. Results

### 4.1. Quantitative Component

A total of 331 participants were enrolled in the study. Of these, 10 participants were removed secondary to missing data. A further four participants were removed as these participants did not meet the eligibility criteria (these participants were under the age of 18 years). This left a total of 317 participants, which formed the sample to determine vaccination coverage among the study population. Of these participants, 21 participants had more than two doses of the vaccine and were up to date with all vaccines. These participants had completed an initial vaccination regimen and all booster vaccine doses required, as described by the national vaccination program mandate circulating at the time of enrolment, and therefore were not eligible for additional vaccinations [27,28]. This left 296 participants eligible for vaccination, either with an initial vaccination, a second vaccine dose, or a booster vaccine.

#### 4.1.1. Vaccination Uptake at Enrolment

Of the entire study population, 296 participants were eligible for a COVID-19 vaccination on the day of enrolment. This included 208 participants who had a previous vaccine through a vaccination regimen and 88 participants who were vaccine naïve. Of all vaccine-eligible participants, 50% (*n* = 148) opted to get a vaccination at the site on the day of enrolment. This number comprised 135 participants (65%) who had a previous vaccination, and 13 participants (15%) who had not previously been vaccinated. It must be noted that 35% (*n* = 73) who had previously been vaccinated (*n* = 208) opted not to have another vaccination on the day of enrolment. Figure 1 describes an overview of the study population and the uptake of vaccination at enrolment. 

#### 4.1.2. Vaccination Coverage and Vaccination hesitancy

Of all participants (*n* = 317), 72% (*n* = 229) had previously received at least one dose of the vaccine. This includes 151 participants (48%) that had at least two doses of the vaccine and 78 participants (25%) with only one dose of the vaccine (incomplete initial regimen with Pfizer or J&J vaccine). Of the participants with an incomplete initial vaccine regimen, 42 participants (13%) took up a same-day vaccine at the vaccination site, thereby completing the initial vaccination regimen with two doses of the vaccine, bringing the total number of participants with two or more doses to 193 participants (61%). This brought the vaccine coverage for those with two doses of the vaccine from 48% to 61% in the total study population, as seen in Figure 2.

The other group included vaccine naïve participants. In total, 88 participants (30%) were vaccine naïve and eligible for a vaccination against COVID-19. Of these 88 participants, 13 participants (15%) opted for a vaccination on the day of enrolment. Vaccination of these previously vaccine naïve participants increased the overall vaccination coverage from 72% to 76% (Figure 3). However, it is noted that 75 participants (24%) of the study population remained vaccine naïve, reflecting vaccine hesitancy amongst this subset of the study population.

For the descriptive and statistical analysis described below, participants were categorized into two groups based on vaccination status. The first group was participants who were vaccinated, comprised of participants having at least one dose of the vaccine (including participants who had a vaccination at the time of enrolment at the vaccination site). The second group comprised participants who were vaccine naïve.

#### 4.1.3. Sociodemographic Information of Participants

The vaccination site, located within a central hospital, was accessible to all persons who attended the hospital. This includes not just the patients attending the hospital for medical-related matters (*n* = 162, 51.1%), but also the companions of these patients (*n* = 80, 25.2%), the staff of the hospital (*n* = 40, 12.6%), and persons of the population who actively sought a COVID-19 vaccination and therefore visited the hospital to seek services from the vaccination site (*n* = 32, 10.1%). Table 1 illustrates a breakdown of the category of participants concerning their reason for attending the hospital and their vaccination status. From this table, the majority of participants within each category had a previous vaccination, with a large proportion of patients (*n* = 122, 75.3%) attending the hospital for medical matters and having been vaccinated with at least one dose.

The participants’ characteristics were further classified as seen in Table 2 below. Participants were described by their socio-demographic characteristics, including age, gender, employment status, and nationality, as well as by their underlying medical conditions, which included hypertension, diabetes, HIV, and other diseases. These were described using descriptive analysis and a chi^2^ test statistic to test the significance of variables to the outcome variable, the vaccination status of the participant.

The mean age of those who were vaccinated versus participants who were vaccine hesitant were similar, at 43.7 years and 43.4 years of age, respectively. However, from Table 2 below, it is evident that older participants were more likely to be vaccinated. Younger participants were more likely to have never been vaccinated. This demonstrates the better predisposition of the older generation to opt for vaccination compared to their younger peers.

Women represented 54% of the study population and were more likely to be vaccinated (*n* = 136, 56.2%) compared to their male counterparts (*n* = 106, 43.8%). The majority of the study population was unemployed (*n* = 202, 64.7%). The unemployed participants who were vaccinated comprised almost two-thirds of the study population (*n* = 152, 62.85) as compared to the participants who were employed and vaccinated (*n* = 90, 37.2%). The majority of participants were South African (*n* = 308, 97.2%). In both subsets of nationality, the majority were vaccinated.

Participants with diabetes accounted for almost ten percent (*n* = 31, 9.7%) of the study population and showed that the majority in this group had been vaccinated (*n* = 25, 80.6%). Thirty-two percent (*n* = 102, 32.2%) of the study population had hypertension, whereas those that had been vaccinated in this group were similarly high (*n* = 86, 84.3%). Participants living with HIV (*n* = 19, 6%) reflected the same high percentage of those who were vaccinated (*n* = 16, 84%). Other chronic diseases accounted for just over a third of the study population (*n* = 111, 35%), and demonstrated a lower vaccination prevalence found in this group (*n* = 79, 71%) as compared to the diabetic, hypertensive, and HIV disease groups.

From all the participants with underlying medical conditions, it is clear that the majority of participants have been vaccinated. A breakdown of other diseases can be described in Table 3. The classification of these diseases was recorded as per the WHO, ICD 11 coding tool [29].

#### 4.1.4. Factors Associated with Vaccination Status

A univariable analysis was conducted to determine the significant factors associated with the outcome of having been vaccinated, as seen in Table 4. From the univariable analysis, significant predictor variables (*p*-value < 0.250) were included in the multiple logistic regression model, using the automatic forward technique, to describe the multivariable analysis. These predictor variables included ‘Age Category’, ‘Hypertension’, and ‘Other Diseases’. Predictor variables deemed clinically significant, but found insignificant in the univariable analysis, were also included in the multivariable analysis. These variables included ‘Diabetes’ and ‘Gender’. In the multivariable analysis, ‘Age Category’ and ‘Other Diseases’ were found to be significant. The age category of 40–64 years had greater odds of being vaccinated compared to the 18–39-year age category (aOR 3.48, 95%CI: 1.85–6.53, *p* = 0.000). The age category of 65–90 years had the highest odds of being vaccinated as compared to the 18–39-year age category (aOR 5.22, 95% CI: 2.13–12.83, *p* = 0.000). ‘Other Disease’ was also found to be significant. Those who did have other diseases had lesser odds of vaccinating compared to those who did not have other diseases (aOR 0.39, 95%CI: 0.21–0.72, *p* = 0.003).

#### 4.1.5. Knowledge of the Vaccination Site

Participants were informed about the vaccination site within the hospital through various mechanisms described in Table 5. From the entire study population, 63.7% (*n* = 202) of participants knew of the vaccination site through specially appointed health promoters aimed to increase awareness of the vaccination site. This was followed by knowledge of the vaccination site acquired through healthcare workers (doctor/nurse), accounting for 15.8% (*n* = 50) of the study population. Visual representation, which included posters throughout the hospital, accounted for 7.3% (*n* = 23). Specific clinical hospital departments (i.e., family medicine, internal medicine, and obstetrics and gynaecology) made participants aware of the vaccination site and accounted for 6% (*n* = 19) of all knowledge. This was followed by community referrals at 4.7% (*n* = 15). Lastly, 1.9% (*n* = 6) of participants knew the vaccination site through other mechanisms (*n* = 6, 1.9%).

#### 4.1.6. Vaccine Hesitancy in Vaccine Naïve Participants

With the convenience of a vaccination site at a central hospital, which is easily accessible to those who visit the hospital, 24% (*n* = 75) of participants remained unvaccinated. An analysis of the reasons not to vaccinate is provided in Table 6 below. Within this group, 30.7% (*n* = 23) did not want to get the vaccine, 28% (*n* = 21) did not believe in the vaccine, 10.7% (*n* = 8) informed the study team that they would come back another day, 9.3% (*n* = 7) did not have the time and 21.3% (*n* = 16) described other reasons for not wanting a vaccine. The other reasons included a concern about the complications or side effects of the vaccine, decreased risk perception of COVID-19 or the decreased need for the vaccine, participants’ requiring a medical specialist’s consultation before vaccination, and vaccine inefficiency. These reasons are further illustrated in Figure 4 below.

### 4.2. Qualitative Findings

Forty-four (44) participants were interviewed in the qualitative component of the study. The interviews were designed to elicit deeper insights into the perceptions of the COVID-19 vaccine, and the acceptability of a vaccination site at a central hospital. Of these 44 participants, 15 chose to vaccinate at the vaccination site on the day of enrolment, and 15 chose not to vaccinate. Out of the total 44 participants, 14 were staff members actively involved in the operations of the vaccination site.

Although inter-related and not mutually exclusive, five major themes emerged from the interviews, which were: perceptions of vulnerability; vaccine safety and efficacy concerns; information gaps and mistrust leading to hesitancy; the value of convenience in the decision to vaccinate; and the role of health promoters.

#### 4.2.1. Perceptions of Vulnerability

Most participants agreed that vaccination is necessary to promote health and prevent illness and death secondary to COVID-19. There was an understanding that vaccines protect everyone and aid in promoting life. This had also been echoed by those opting not to vaccinate.

“It helps with COVID-19; so that one does not get infected with COVID.”(Key Informant 19)

Some participants felt that there was no need to vaccinate or get a booster, reflecting a perception of low risk of illness or death from COVID-19.

“It’s not that I don’t want to be vaccinated in your site, the thing is I have already gotten an injection already, so I no longer see the importance of getting the booster.”(Key Informant 10)

Staff felt anecdotally that mainly older individuals chose to vaccinate, with females having a slightly greater inclination to vaccinate than their male counterparts. This suggests that the perception of vulnerability to COVID-19 varies depending on factors such as age and gender.

#### 4.2.2. Vaccine Safety and Efficacy Concerns

Many participants described concerns related to the COVID-19 vaccine. Some of these concerns were related to the side effect profile, in that participants were afraid of complications, including long-term morbidity and death associated with the vaccine, as well as the vaccine’s interaction with their chronic medication. Some participants did not believe in the efficacy of the vaccine in offering protection against COVID-19 at all, while one participant went as far as to describe the vaccine as a devilish entity. General misconceptions, lack of information about the vaccine, and mistrust were identified by the hospital staff participants as deterrents to vaccination for the population.

“I want to ask my doctor first about it is safe to do it or it’s not safe. Because if I can do it, and then have complications, the doctor he will say that I didn’t tell you to do it. So, I must ask him first that’s when I will come back and get it.”(Key Informant 12)

“I’m not saying that the vaccine is not helping people, it is. But me personally—like I said—yeah, I’m worried about it affecting my diabetic treatment.”(Key Informant 18)

“I’m scared it’s going to be the 666 and I don’t want to go to hell, I want to go to heaven.”(Key Informant 27)

#### 4.2.3. Information Gaps and Mistrust Leading to Hesitancy

There was a sense from some of the participants that there is not enough evidence and research available on the vaccine, and that more information is required. Some participants wanted to obtain information from their healthcare providers or do their own reviews on the vaccine and its safety.

“I personally feel that the amount of research that went into the vaccine was not enough, the timeframe was very short since the announcement of the COVID pandemic and the vaccine coming out.”(Key Informant 35)

Additionally, the hospital staff interviews identified that there were divergent views among healthcare workers themselves on vaccination, with some outright opposing the availability of the vaccination site and the vaccine at the hospital and influencing patients against it. This illustrates a level of vaccine hesitancy among healthcare workers themselves, at the hospital.

“Healthcare workers have been very resistant to coming in for vaccination and to a large extent, they influenced the patient, the caregivers, and the patient’s opinion about vaccination.(Key Informant 35)

#### 4.2.4. Value of Convenience in Decisions to Vaccinate

A consensus among the majority of participants was that the vaccination site was convenient in its location. Participants described that the site in relation to the pharmacy and the outpatient clinics allowed for one to engage in multiple healthcare services at one visit. The site itself was described as having friendly professional staff, with minimal waiting times, and allowed for social distancing. Participants also felt at ease knowing the vaccination site was in a hospital and in close relation to hospital resources, including doctors and nurses.

“It is easy to come to the hospital to fetch the medication and to get vaccinated.”(Key Informant 26)

“What I can say is that maybe it’s a little bit convenient [speaking of the vaccination site]. It’s a little bit more convenient than visiting an outside place.”(Key Informant 38)

#### 4.2.5. The Role of Health Promoters

Health promoters were echoed, amongst all participants, to be a huge influence on the education of the vaccine and encouraging vaccination. The majority of participants described that they had been vaccinated or that they became aware of the vaccination site through interaction with a health promoter or staff member of the vaccination site.

“I was about to go home (after) finishing collecting my medication, so I met the sister (nurse) there, (s)he told me about the vaccination, that there is a vaccination site here at the hospital.”(Key Informant 24)

## 5. Discussion

The establishment of a vaccination site at this central hospital in South Africa enabled the study to assess vaccination coverage, uptake at enrolment, factors influencing uptake, and perceptions and acceptability of both vaccination and the site itself.

The vaccination coverage (at least one dose) of the study population, before offering a vaccination at enrolment, was 72%. This increased by only 4%, to 76%, after participants were offered a vaccination at the vaccination site at the time of enrolment. The vaccination coverage found within the study population is in line with the global target for vaccination coverage of 70%, as determined by the World Health Organization [30]. This vaccination coverage is also higher than the target vaccination coverage of 67% stipulated by the Department of Health for South Africa, to have been achieved by the end of 2021 [7]. It must be noted that vaccination coverage within this study population is different from South Africa’s national vaccination coverage. As of February 2024, South Africa had a vaccination coverage of 44.9% of the adult population (18 years or older), where this population had received at least one dose of a COVID-19 vaccine [15]. This is similar to the vaccination coverage of 44.6% demonstrated in Gauteng province in the adult population [15]. Differences in these statistics could highlight that, due to the study being conducted in a hospital within an urban setting, vaccinations were more prevalent amongst those attending the hospital than the national population, resulting in higher vaccination coverage.

Having a vaccination site at a central hospital did not have a notable impact on total vaccination coverage (at least one dose), only increasing vaccination coverage of the study population by 4%. However, the convenience and accessibility saw the number of participants with two or more doses of the vaccine increase from 48% to 61%. The vaccination site also saw 50% of eligible participants participate in the uptake of a same-day vaccination. Only a few participants who were vaccine naïve engaged in vaccination uptake at enrolment, as compared to the majority of participants who had at least one dose of the vaccine. The majority of the vaccine naïve participants remained vaccine-hesitant. Therefore, understanding factors around vaccination is necessary to determine the acceptability of the vaccination at this site.

Multiple factors were identified that influenced a participant’s decision to vaccinate. These factors included socio-demographic features such as age and gender; comorbidity status, perceptions of vulnerability, vaccine safety and efficacy concerns, information gaps and mistrust surrounding the vaccine, and convenience and accessibility in acquiring vaccination.

This study showed age as the most important statistically significant factor associated with vaccination uptake. This was echoed among participants, especially among staff within the vaccination site. This factor links in with older adults’ perceptions of their own vulnerability to COVID-19 infection and the acceptance that the COVID-19 vaccine plays a role in promoting their health and well-being. Younger individuals, within the age group of 18–39 years, were found to be less likely to vaccinate than their older counterparts, suggesting a decreased perception of COVID-19 susceptibility. This is comparable to a study conducted in Egypt that concluded that the age groups 18–29 and 30–39 years were associated with a low vaccination uptake [30]. A global survey further resonated with our findings, stating older adults were more likely to vaccinate [31]. However, these findings are contradictory to a study conducted in Italy that investigated the reasons for receiving a second COVID-19 booster in adults and people with chronic medical conditions. That study revealed that younger participants decided to receive the second booster dose due to their risk of acquiring the disease and thereby transmitting it to others [32]. Contradictory findings across the countries could be explained by differing perceptions of vulnerability between the geographical populations.

The female gender represented a greater proportion of those who were vaccinated as compared to their male peers, reflecting an increased perception of vulnerability to COVID-19 and a decreased hesitancy towards vaccination. Similar findings were documented in Ghana, as well as in a global survey, where older women showed less vaccine hesitancy than males [31,33]. In contrast, a systematic review conducted by the University of Wellington in New Zealand found that males were more likely to get vaccinated than females [18]. However, many COVID-19 observational studies had an overrepresentation of women, due to the vast majority of studies being conducted in healthcare workers, where many were female. This imbalance reflects that there are no gender-specific recruitment strategies in trials [34]. In our study, women represented 55% of the study population, therefore slightly overrepresented. Even with these contradicting findings, more emphasis is needed to increase vaccination uptake in the male population as being male is a significant mortality-related risk factor for COVID-19, as shown by Dessie, Z et al. [25].

This global systematic review also highlighted a higher in-hospital mortality risk for those with diabetes and hypertension [25]. HIV has also been identified as a risk factor for COVID-19 mortality [35]. From our study, vaccine uptake was greater in those with these diseases. This would suggest that those with co-morbidities view themselves at risk of acquiring COVID-19 infection and complications. This was echoed in the study by Miraglia et al., where participants with at least one chronic medical condition perceived themselves at greater risk for severe infection of COVID-19, and therefore were more likely to receive a COVID-19 second booster dose [32]. Contrastingly, our study also illustrated that those with comorbidities were also concerned about possible adverse effects of the vaccine on their chronic disease state and potential interactions with their medication. This is consistent with the findings of an Indonesian study describing similar fears that accounted for a lower uptake of booster vaccinations [19].

The pregnant population has also been described to be at a greater risk for developing severe and complicated COVID-19 infection. Therefore, the pregnant population stands to benefit from the COVID-19 vaccine. However, it has been described that vaccination rates among this group are lower than expected in a high-risk population [36]. The major contributing factor to vaccine hesitancy in this population is concern for the safety of the unborn child [37]. Pregnant women were not covered in our study; however, this provides an opportunity for future research in this area.

Apart from the fears described above, many psychological factors negatively impact vaccine uptake including the fast development of the vaccine, safety, and negative side effects [18]. These sentiments are further echoed in a qualitative study where common ideas that emerged as inhibiting factors included, the short development duration of vaccines, the COVID-19 vaccine side effects, and death after vaccination [19]. This was further reiterated by a study conducted in China to determine the acceptance of COVID-19 vaccination in different COVID-19 epidemics. In this study, it was found that the acceptance of an immediate vaccination had declined, and this was due to concerns about vaccine safety [20].

The fears around vaccination, side effect profile, efficacy, and safety can largely be attributed to information gaps surrounding vaccines. As shown in a study conducted in Zambia, participants were more likely to accept the vaccine if they were better informed on the vaccine [38]. Wang et al. documented that a doctor’s recommendation was an important factor in the decision to vaccinate, with an odds ratio of 3.13 in favour of immediate vaccination [20]. Further to this, a study in Italy found that physician information on COVID-19 vaccination was significant in parents’ decision to have their children vaccinated [39]. This study, along with other studies, demonstrates that healthcare workers are a trusted source of information for participants, and play a role in participants’ decision to vaccinate [19,38,40].

The qualitative component of this study demonstrates that there exists a level of vaccine hesitancy among healthcare workers, which influences patients’ decision to refuse vaccination. Vaccine hesitancy among healthcare workers has been documented elsewhere, resulting in advice to patients not to vaccinate [40]. This finding proves to be of importance, as addressing healthcare worker knowledge is a key factor for increased vaccination uptake in both healthcare workers and patients [40,41].

Concerning the location of the vaccination site, a prominent factor that influenced vaccination uptake was the concept of convenience with respect to the location of the vaccination site within the central hospital. It allows for easy access, especially for those patients with multi-morbidities and for staff working in the hospital. In our study, the location of the vaccination site was convenient, such that multiple healthcare services could be provided during a single visit to the hospital. A study conducted in Cyprus on healthcare workers’ vaccination uptake mirrored the same finding that easy access to a vaccine site influenced the decision to vaccinate [41].

A small proportion of patients offered the vaccination at the hospital site requested time to think about it. One of the downsides of offering the service on the same day as the routine hospital visit for patients that only visit the hospital monthly, or even every six months, is that it does not cater to the group that needs more time to decide to vaccinate, after reflecting on the choices or taking advice from peers or family members. A similar sentiment was found in a study conducted in Canada. To understand vaccine decision-making, this study documented that many who opted to delay their vaccinations demonstrated hesitancy and concerns over the vaccines [24]. However, this study revealed that there was an increased willingness to vaccinate as time passed in this group, using the time that passed to determine evidence of safety, efficacy, or uncertainty as a whole [24]. Therefore, uptake of the vaccine on the same day may be limited in the group that needs more time for a decision to vaccinate. However, the location of the site with respect to their hospital visits will allow them to access a vaccine at their next visit if they express willingness to vaccinate at a later stage.

The time taken to decide to vaccinate is crucial; however, the time spent at the vaccination site may also pose a barrier to one receiving a vaccine [42]. A study conducted in the United Kingdom demonstrated that persons who were faced with busy schedules, with limited general practitioner visit options, found it difficult to prioritize obtaining a vaccination [43]. To address the limited time availability of our participants, their prescriptions were fast-tracked through the pharmacy queue while they were administered their vaccinations. During this time spent at the vaccination site, participants were offered coffee as well as a muffin. Having this strategy at the vaccination site encourages people to obtain a vaccine, knowing that they would still acquire their medication and not delay their time spent at the hospital.

The research project conducted had potential methodological limitations. Firstly, the study was a cross-sectional study, therefore restricting the ability to make temporal relationships between the independent and dependent variables. Secondly, the study was conducted in one hospital facility, consequently resulting in a sample population that may not be representative of the country’s population as a whole. For this reason, the findings of this study should be interpreted carefully as they may not be generalized to the greater population, specifically those with chronic medical conditions in the South African context. Thirdly, the self-administered questionnaire may have been subjected to non-response bias and social desirability bias. Extensive steps were taken to minimize these biases, including extensive counseling during informed consent and stressing to participants the value of capturing their perceptions, both those vaccinating and non-vaccinating. Notwithstanding this, our study has several strengths. Firstly, the use of both quantitative and qualitative methods enabled us to capture factors that influence the decision to vaccinate and to measure vaccine acceptability. In the results, we show that the qualitative findings confirmed the findings of the quantitative component, therefore strengthening the value of our findings.

## 6. Conclusions

This study has shown that it is logistically feasible and acceptable to provide a vaccination site at a large hospital targeting patients attending outpatient services for chronic medical conditions and that such a service also benefits accompanying persons and hospital staff.

Whilst the need was less than anticipated, with more than seventy percent of participants already having at least a single vaccination dose, the convenience of a hospital-based vaccination site resulted in 50% of participants’ uptake of a same-day vaccination.

The intervention did not significantly impact vaccination coverage in this population, defined as having at least one vaccination dose, but significantly increased the number of participants who were vaccinated twice or more from a prior level of 48% to 61%. Vaccine hesitancy was high (24%) in a population that perhaps has the greatest need for this medical countermeasure. Providing the service on-site at a hospital, where vaccine-hesitant persons had access to direct information about the importance of vaccinating by health promoters and their own medical and nursing caregivers, did not address this phenomenon.

Addressing vaccine hesitancy in the populations that are of greatest need, in the general public, and amongst healthcare workers is essential to ensure higher uptake of vaccinations in the future.

## Figures and Tables

**Figure 1 diseases-12-00113-f001:**
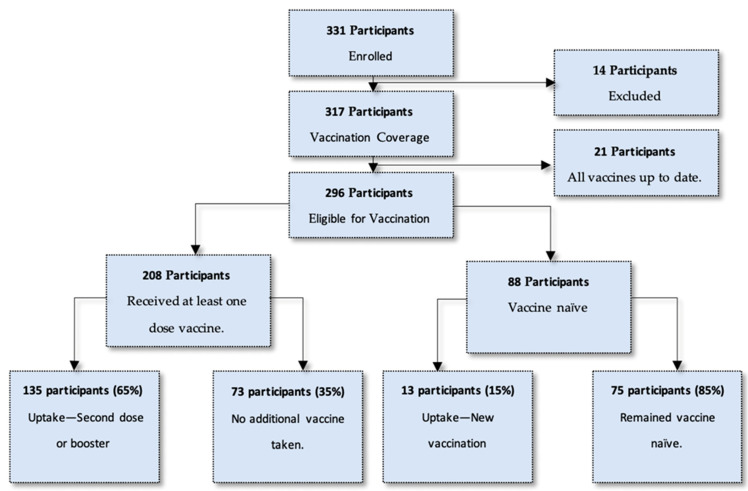
Breakdown of study population and uptake of vaccination at enrolment.

**Figure 2 diseases-12-00113-f002:**
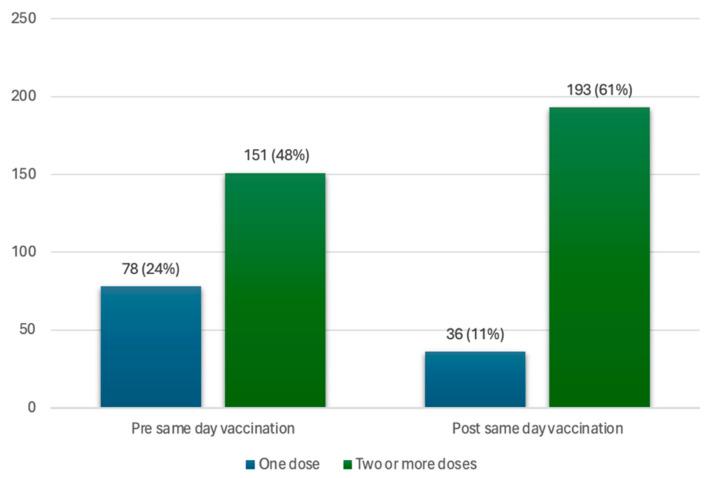
Coverage of one dose and two or more doses of the vaccine, in the vaccinated population, pre- and post-same-day vaccination uptake.

**Figure 3 diseases-12-00113-f003:**
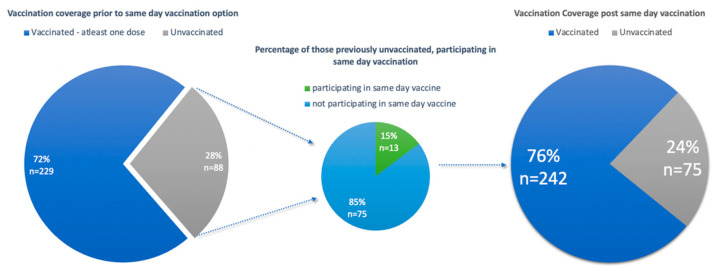
Graphs illustrating vaccine coverage of the study population pre- and post-same-day vaccination.

**Figure 4 diseases-12-00113-f004:**
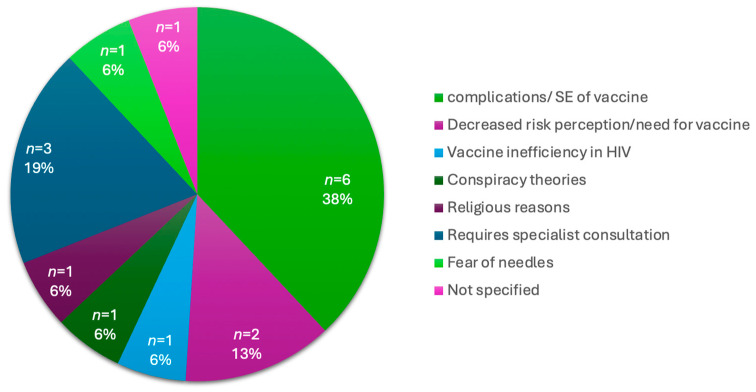
Breakdown of “Other Reasons” for vaccine hesitancy in vaccine naïve participants.

**Table 1 diseases-12-00113-t001:** Category of participants attending the hospital.

Category of Participants	Number (N)—Vaccinated (%)	Number (N)—Vaccine Naïve (%)	Total
**Patient**	**122 (75.3)**	**40 (25.7)**	**162**
Hospital Visit	59	34	69
Medication Collection	63	6	93
**Patient Companion**	**58 (72.5)**	**22 (27.5)**	**80**
Accompanied Patient for Visit	43	17	60
Mediation Collection on Behalf of Patient	8	4	12
Visiting Inpatient	7	1	8
**Staff**	**28 (70)**	**12 (30)**	**40**
**Community members—Attending** **for Vaccination**	**32 (100)**	**0 (0)**	**32**
**Other**	**2 (66.6) ***	**1 (33.3) ****	**3**
**Total**	**242 (76.3)**	**75 (23.7)**	**317**

* 1st participant—CV submission; 2nd participant—collected documents from the hospital. ** Student attending hospital for practices.

**Table 2 diseases-12-00113-t002:** Characteristics of participants described by participants who vaccinated and participants who were vaccine naïve.

Characteristics	Number Vaccinated out of 242 (*n*)	Percentage of Those Vaccinated (%)	Number Vaccine Naïve out of 75 (*n*)	Percentage of Those Vaccine naïve (%)	Total Participants out of 317 (N)	*p*-Value
**Socio-demographic**						
**Age Category**						
18–39	83	34.3	45	60.0	128	0.000
40–64	109	45.0	22	29.3	131	
65–90	50	20.7	8	10.1	58	
**Gender**						
Male	106	43.8	38	50.7	144	0.297
Female	136	56.2	37	49.3	173	
**Employment Status**						
Employed	90	37.2	25	33.3	115	0.544
Unemployed	152	62.8	50	66.7	202	
**Nationality**						
South African	236	97.5	72	96.0	308	0.488
Non-South African	6	2.5	3	4.0	9	
**Underlying medical** **condition**						
**Diabetes**						
Yes	25	10.3	6	8.0	31	0.553
No	217	89.7	69	92.0	286	
**Hypertension**						
Yes	86	35.5	16	21.3	102	0.021
No	156	64.5	59	78.7	215	
**HIV**						
Yes	16	6.6	3	4.0	19	0.405
No	226	93.4	72	96.0	298	
**Other**						
Yes	79	32.6	32	42.7	111	0.112

**Table 3 diseases-12-00113-t003:** Breakdown of other diseases as classified by the ICD-11 codes—WHO.

Category of Other Diseases	Number (*n*)—Vaccinated	Number (*n*)—Vaccine Naïve	Total
Neoplasms	4	1	5
Diseases of circulatory systems	24	16	40
Diseases of digestive system	4	2	6
Diseases of blood or blood forming organs	1	0	1
Diseases of immune systems	1	1	2
Diseases of endocrine and nutrition	7	1	8
Diseases of genitourinary systems	2	0	2
Diseases of infectious and parasitic	2	1	3
Diseases of mental, behavioural	4	1	5
Diseases of musculoskeletal	16	4	20
Diseases of nervous systems	4	2	6
Diseases of respiratory system	7	2	9
Diseases of skin	2	0	2
Diseases of visual systems	1	0	1
Other	0	1	0
Total	79	32	111

**Table 4 diseases-12-00113-t004:** Univariable and multivariable analysis of factors associated with vaccination status.

**Univariable Analysis**			
**Characteristics**	**Crude OR**	**95% CI**	** *p* ** **-Value**
**Socio-demographic**			
**Age category (years)**			
18–39	1.00		
40–64	2.69	1.50–4.82	0.001
65–90	3.38	1.48–7.77	0.004
**Gender**			
Male	1.00		
Female	1.38	0.78–2.21	0.298
**Employment status**			
Unemployed	1.00		
Employed	1.18	0.69–2.04	0.544
**Nationality**			
South African	1.00		
Non-South African	0.61	0.15–2.50	0.493
**Underlying medical condition**			
**Diabetes**			
Yes	1.32	0.52–3.36	0.554
No	1.00		
**Hypertension**			
Yes	2.03	1.10–3.74	0.023
No	1.00		
**HIV**			
Yes	1.69	0.48–6.00	0.410
No	1.00		
**Other Disease**			
Yes	0.68	0.40-1.15	0.148
No	1.00		
**Multivariable analysis**			
**Characteristic**	**Adjusted OR**	**95% CI**	** *p* ** **-Value**
**Age category (years)**			
18–39	1.00		
40–64	3.48	1.85–6.53	0.000
65–90	5.22	2.13–12.83	0.000
**Other Disease**			
Yes	0.39	0.21–0.72	0.003
No	1.00		

**Table 5 diseases-12-00113-t005:** Knowledge of the vaccination site.

Characteristics	Number (*n*)	Percentage of Total Participants *n* = 317 (%)
**Knowledge of vaccination site**		
Doctor/Nurse	50	15.8
Overheard another patient	2	0.6
Visual	23	7.3
Community	15	4.7
Healthcare promoter	202	63.7
Hospital department	19	6.0
Other	6	1.9
Total	317	100

**Table 6 diseases-12-00113-t006:** Reasons for vaccine hesitancy in vaccine-naïve participants.

Reasons for Vaccine Hesitancy in Vaccine Naïve Participants	Number (*n*)	Percentage of Total Participants *n* = 75 (%)
Do not believe in vaccines	21	28
Do not have the time	7	9.3
Do not want to get vaccine	23	30.7
Will come back another time	8	10.7
Other	16	21.3
Total	75	100

## Data Availability

Data supporting reported results can be requested from the authors.

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
