# Peer review of "Feasibility of Provision and Vaccine Hesitancy at a Central Hospital COVID-19 Vaccination Site in South Africa after Four Waves of the Pandemic"

_diseases, 2024, doi:10.3390/diseases12060113_

Round 1

Reviewer 1 Report

Comments and Suggestions for Authors

Introduction

a)       The Authors should discuss the vaccine hesitancy since it gives a significant contribution to the knowledge, attitudes, and practices and to the suboptimal vaccination coverage.

Methods

a)       It is stated that the study was conducted in one hospital but it is necessary to clarify how the hospital has been selected and how many are present in the geographic area.

b)      It is not indicated how the hesitancy has been measured.

c)       It is necessary to specify when the data collection was done.

d)      It is necessary to clarify how the informed consent has been obtained from all participants before the commencement of the study.

e)       It is not stated whether the participant was informed about the use and anonymization of the data and that survey responses guarantee the anonymity of each participant.

f)       It is not given any information whether participants were not able to continue to the next question of the questionnaire if they failed to provide a response to an item.

g)      It should be clarified whether the participants have received any gift or monetarily compensated.

h)      The Authors should describe the survey questionnaire items.

i)        It should be clarified whether a pilot study has been conducted.

j)        The Authors should clarify about the face-validity testing of the questions with an explanation of the validity of the content of the questions with regard to the research aims. The Authors should clarify how they had estimated the reliability, or internal consistency, of the questions by using, for example the Cronbach’s alpha in order to measure whether or not a score is reliable.

k)      One of major weakness is that univariate and multivariate statistical analysis has not been performed.

Results

a)     It is not reported the response rate.

b)    Was there any attempt to quantify the response bias: no information about the non-responders has been reported. It would be useful to have some kind of indication of comparability with non-respondents. Is there any population-based data available? How did they differ from those in the sample, how representative is the sample and were the findings representative of the country?

Discussion

a) One weakness is that the results are not compared with other studies conducted in other countries. The work should therefore be enriched in such a way as to become self-supporting by photographing the context and what is around it in order to make comparisons. The authors should expand their comments regarding the results of the study and the literature review by adding a comparison with the results of previous surveys conducted in different geographic areas. For example, the following article should be cited Miraglia del Giudice et al. 2023;11(4):737; Liu et al. Vaccines (Basel) 2021;9(3):29; Fisher et al. Ann Intern Med 2020;173(12):964-73.

b) The pivotal role of healthcare providers as source of information with a positive impact towards vaccination should be stressed and studies supporting this statement should be added. For example, the following articles should be cited Miraglia del Giudice et al. Vaccines 2022, 10(3), 396; Wang et al. Vaccines (Basel) 2021;9(3):29.

c) A paragraph regarding the main limitations of the study should be added and discuss all limits such as, for example, the study design, the representativeness of the sample, the generalizability of the results, the recall bias, and the social desirability bias.

References

a)    The manuscript is not well referenced. The References list is not updated, since several recent articles on this topic published on peer-reviewed journals have been not included.

Comments on the Quality of English Language

The paper needs editing by a native English person.

Author Response

Good day 

Thank you for the opportunity to respond to the reviewers’ comments. The responses to the comments are provided on the word document attached titled "27-04-24 Response to reviewer 1_Nair et al". 

Please let me know if you have any further queries or concerns. 

Kind Regards 

Dr Shanal Nair 

Reviewer 2 Report

Comments and Suggestions for Authors

This is an original investigation of COVID-19 vaccination uptake, coverage and hesitancy in a cohort of 317 persons attending a single hospital in Pretoria, South Africa, utilizing a mixed methods research plan. The manuscript is well-organized, well-written and easily understood. The 4 figures and 6 tables are well-constructed, easily read, and both they and their legends complement the data and text. The experimental study design is valid, & data analysis appears to be appropriate and correct. Both the Introduction and Discussion are comprehensive and define the research study questions, goals, and conclusions. The Abstract succinctly defines the manuscript. The references are comprehensive and current with the latest from 2024 (see suggestions for revision).

The authors have prepared a superbly constructed and organized manuscript describing the results of an important study which should prove to be an important contribution to this topic. 

The authors may wish to consider the following comments and suggestions in revising their manuscript.

1) Pregnancy is considered to be an important risk factor for poor clinical outcomes due to COVID-19. Although it is not a focus of this investigation, perhaps it should be mentioned as a risk factor in the Discussion for the sake of completeness. In addition, you might wish to cite a new 2024 article in the Yale Journal of Biology and Medicine dealing with COVID-19 vaccine hesitancy among pregnant women: doi.org/10.59249/LPOQ5146 

2) There were 173 female study patients in this cohort. Were any pregnant? Can you state that in your paper?

Author Response

Good day 

Thank you for the opportunity to respond to the reviewers’ comments. The responses to the comments are provided on the word document attached titled "27-04-24 Response to reviewer 2_Nair et al". 

Please let me know if you have any further queries or concerns. 

Kind Regards 

Dr Shanal Nair 

Round 2

Reviewer 1 Report

Comments and Suggestions for Authors

As already indicated in my previous revision, all main limitations of the study should discussed such as, for example, the study design, the representativeness of the sample, the generalizability of the results, and the recall bias.

Comments on the Quality of English Language

Minor editing of English language are required.

Author Response

Good day 

Thank you for the opportunity to respond to the reviewer's comments. The responses to the comments are provided in the document attached.

Please inform if any further information or clarifications are required. 

Kind Regards 

Shanal 
